# Microbial Carbon Metabolic Functions in Sediments Influenced by Resuspension Event

**Miao Wu** [1], **Ming Zhang** [2], **Wei Ding** [1], **Lin Lan** [2], **Zhilin Liu** [1], **Lingzhan Miao** [1] and **Jun Hou** [1,*]

1   Key Laboratory of Integrated Regulation and Resources Development on Shallow Lakes, Ministry of Education, College of Environment, Hohai University, Xikang Road 1st, Nanjing 210098, China; wumiao@hhu.edu.cn (M.W.); dwhhu.cn@hhu.edu.cn (W.D.); lzl1993@hhu.edu.cn (Z.L.); lzmiao@hhu.edu.cn (L.M.)

2   Jiangsu Province Water Resources Planning Bureau, Nanjing 210098, China; zhangming20201013@163.com (M.Z.); hugo_lan@126.com (L.L.)

*   Correspondence: hjy_hj@hhu.edu.cn; Tel./Fax: +86-25-83787930

**Abstract:** Microorganisms in sediments are an important part of the aquatic ecosystem, and their functional activities are sensitive to external environmental pressure. Shallow lakes are characterized by frequent sediment resuspension events, leading to large amounts of nutrients being released. However, information about the potential impacts of sediment resuspension events on the functional activities of microbial communities is limited. In this study, the responses of microbial carbon metabolism in sediments under different wind–wave disturbance were analyzed by BIOLOG ECO microplates. The results showed that under four disturbance conditions (wind speeds of 0, 1.60, 3.62, and 14.10 m/s), the total carbon metabolism function of the sediment microbial community (represented as average well-color development, AWCD) remained unchanged ($p > 0.05$), and the final total AWCD value stabilized at about 1.70. However, compared with the control group, some specific carbon sources (e.g., amines and carboxylic acids) showed significant changes ($p < 0.05$). We found that short-term (8 h) resuspension events did not affect the total carbon metabolism of sediment microbial communities, while it affected the microbial utilization ability of some specific types of carbon sources. For example, we found that the microbial utilization capacity of polymers in the 14.10 m/s group was the best. This study provides a new insight into the carbon cycle process of shallow lake sediments that resuspension events will affect the carbon cycle process of sediments.

**Keywords:** sediment; BIOLOG ECO microplate; metabolic functions; resuspension events; AWCD



## 1. Introduction

Sediments are an important part of aquatic environments, and are involved in biogeochemical cycles and energy transformation. Resuspension of sediment events are very common in natural water systems, including rivers, estuaries [1], intertidal zones [2], shallow lakes [3,4], and shallow sea areas [5]. Studies have confirmed that the surface sediment is easily resuspended by the dynamic action of wind–waves [6], tides [7,8], storms [9], and human disturbance [10–12]. The resuspended sediment entering the water body usually contains a large number of microorganisms, biological debris, residues, granular phosphorus, and other nutrients. Consequently, the resuspension event process will inevitably cause material exchange between the water body and the suspended matters. Numerous studies have shown that the process of sediment resuspension events can promote the release of nutrients [13,14]. Qin explored nutrient release models in sediments under static and dynamic conditions [15]; and found that the disturbance time could affect phosphorus transformation in sediments [16]. Tang studied the dynamic characteristics of sediments and nutrients in Taihu Lake under various natural disturbances [17], and the results suggested that both wind speed and wind fetch length could effectively activate the sediment layer, and trigger particles entrainment into the overlying water. In addi-

tion, sediment resuspension events could change the structure and diversity of planktonic bacterial communities [18]. It has been reported that the dynamics of phytoplankton and zooplankton community structure were influenced by resuspension of sediment events, and different conditions excited distinct roles [19,20].

The sediment consists of complex aggregates of microorganisms, including bacteria, archaea, algae, fungi, protozoa, and metazoans, which play a vital role in the primary production and biogeochemical cycle of freshwater ecosystems [21,22]. Microorganisms in lake sediments play an irreplaceable role in the carbon cycle [23,24], nitrogen cycle [25,26], and biodegradation [27]. For example, there is a thin layer in the sediments (from several millimeters to several centimeters), in which anaerobic processes dominate, with the participation of bacteria. The breakdown of organic matter by bacteria is usually the primary mechanism for feeding an internal reservoir with nutrients [28]. After exceeding the permissible load for a specific tank nutrient, the so-called process of supply or internal import, consisting in the release of nutrients, especially accumulated phosphates in bottom sediments [29,30], takes place. Methane metabolism has an impact on global climate change, and the release of methane in lakes is the result of methanogenesis and methane oxidation, which cannot be completed without the participation of methanogens and methanotrophs, and other microorganisms. The utilization of microbial carbon sources can represent the metabolic functional characteristics and community structure of microorganisms. However, microbial functional activity is sensitive to, and easily affected by, changes in external environmental pressures. A previous study has demonstrated that different aquatic conditions might affect the microbial community structure of biofilms on plastic and natural substrates [31]. Therefore, in order to understand the effects of microorganisms in the sediments during the resuspension event, it is essential to analyze the metabolic functions of the microbial communities, which could also help to reveal their potential impacts on the carbon cycle. In recent years, the BIOLOG ECO microplate has been favored by many researchers due to its simple operation, abundant data, and intuitive reflection of the overall activity of the microbial population [32].

Shallow lakes are characterized by frequent sediment resuspension events, causing the release of large amounts of nutrients. Nevertheless, there have been few studies about the potential impacts of sediment resuspension events on the functional activities of microbial communities [33,34]. Based on the above, an indoor annular flume was used in this study to simulate the sediment resuspension event process. The water characteristics during the sediment resuspension events were monitored. After the resuspension event experiments, a BIOLOG ECO microplate was used to test the response of sediment microbial carbon metabolism. The correlation between the metabolic function of the microorganisms in the surface sediment and the resuspension event process was explored to provide a theoretical basis for the management of aquatic ecological environments.

## 2. Materials and Methods

### 2.1. Materials and Experiments

Lake Taihu, China's third-largest freshwater lake, is situated in the lower Yangtze River Delta. It has a surface area of 2338 $km^2$, and a mean depth of 1.9 m. The water and sediment samples used in the experiment were collected from Gonghu Bay of Lake Taihu in October 2019. The water samples were taken first, and stored at 4 °C. The surface sediment (0–10 cm) was sampled by a Peterson Mud Extractor (WHL 15-HL-CN, Yaou, Beijing, China) which was deployed from a boat, and collected samples from the water sampling point at the bottom of the lake. Next, the sediment samples were placed in sterile sample boxes and brought back to the laboratory within 24 h. Based on our previous investigations, and other studies, the sediment grain size distribution of Lake Taihu is as follows: <5 μm 2.4–14.7%; 5–50 μm, 53.1–82.6%; 50–200 μm, 15.0–34.3% [35–37].

Wind speed is a key factor affecting the movement of water in shallow lakes. In this study, sediment resuspension events induced by wind–wave disturbance were simulated in four indoor pneumatic annular flumes (Figure 1), which is a patented product of Hohai

University (Patent No. 200710025671.3). The experiment was conducted in a biological greenhouse, in which the temperature was $18 \pm 2$ °C. In natural conditions, shallow lakes respond greatly to wind–wave disturbances, which will cause significant hydrodynamic changes in the lakes, and thus resulting in re-suspension affecting material circulation. In order to investigate the effects of resuspension events, many researchers have designed simulated resuspension experiments based on indoor flumes [38–41]. Based on a previous study [42] and the wind speeds in Taihu (range from 0 to 20 m/s), four wind speeds were used in this experiment. Air blasting devices (POPULA, Foshan, China) were used to simulate the wind disturbance, and the wind speeds, measured by acoustic doppler velocity (ADV) software V1.0 from small to large, were 0, 1.60, 3.62, and 14.10 m/s, which were recorded as G1, G2, G3, and G4, respectively. First, a 20 cm thick mud sample was laid in the annular flume, then the in-situ water sample (about 40 cm deep) was slowly injected. Then this system was allowed to stand for two weeks to make it stable, representing the actual sediment aggregates in the lake. After that, the air blasting device was started, to obtain the four wind speeds. After the stabilization (2 h) of flow velocity (set as the beginning), a water sample was collected using a siphon from the water column at 10 cm above the surface sediment at 0, 0.5, 1, 2, 3, 5, and 8 h. After the suspended sediments settled, the surface sediments were collected using an infector, and stored at 4 °C for later use. It is noted that although the disturbance can be regulated by the air blasting devices, it can only be maintained at a constant wind speed, which means it cannot completely simulate the wind–wave disturbance in natural conditions, because the wind speed and direction are constantly changing in natural conditions.

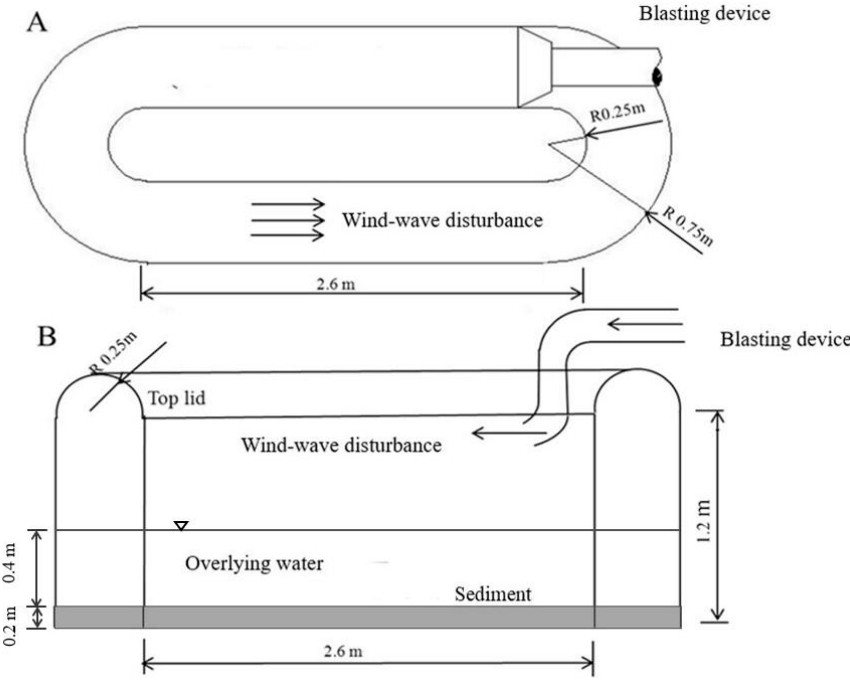

**Figure 1.** Schematic diagram of the annular flume. (**A**) is the planform of the flume, and (**B**) is the side view of the flume.

## 2.2. Determination of Water Characteristics

The collected samples were immediately processed and analyzed. The total nitrogen (TN) and total phosphorus (TP) were measured by ultraviolet spectrophotometry and the molybdenum antimony photometric method, the chemical oxygen demand (COD) was detected by the potassium permanganate oxidation method, and suspended solids (SS) were detected by the gravimetric method, respectively [43].

### 2.3. BIOLOG ECO Microplate Test Method

Resuspension of sediment events will certainly bring changes to the aquatic environment, which will affect the microbial functional activities of microorganisms. To monitor the changes of microbial activity brought by sediment resuspension, several alternative methods can be used, such as enzyme activity assay, BIOLOG ECO Microplate, functional genome sequencing, and so on. In this study, the carbon metabolism characteristics and the functional diversity were determined by BIOLOG ECO Microplate (Hayward, CA, USA) [44], which is convenient and commercially available, and has been widely used in soil [45,46] and sediment [47,48] microbial research with the absence of accurate in situ measurement techniques. The BIOLOG ECO microplate is composed of 31 different single carbon sources, which can be divided into six categories: phenolic acids, carbohydrates, polymers, carboxylic acids, amino acids, and amines/amides [32]. Electrons are produced during the microbial oxidization of carbon sources in microbes. During the measurements, these electrons could be preferentially captured by tetrazolium violet dye, and then changed the color agent to purple [49].

The prepared sediment sample (0.5 g) and sterilized water were added to a triangular flask. The suspension was mixed, and then the slurry was gradually diluted with physiological saline until the absorbance value reached 0.05 at 590 nm. Next, the suspension was added into the wells (150 μL) of the BIOLOG ECO microplate, and incubated at 25 °C for 7 days [50]. During the incubation, the absorbance value was determined at 590 every 12 h using a multifunctional enzyme label tester (Bio Tek, Winooski, VT, USA) for 7 to 10 days, and based on previous studies [51–54].

### 2.4. Determination of Average Well-Color Development Values

The average well-color development (AWCD) was measured to represent metabolic proficiency. The AWCD was calculated as Equation (1).

$$AWCD = \sum_{i=1}^{n}(C_i - R)/n \tag{1}$$

where $C_i$ is the absorbance value of each well at 590 nm, R is the absorbance value of the blank control well, and n is representative of the number of wells. Additionally, a value of $(C_i - R)$ less than 0.06 is counted as zero.

### 2.5. Calculation of Metabolic Functional Diversity Indices

The metabolic functional diversity was represented as the Shannon–Wiener diversity index (H′), Simpson diversity index (D), and Shannon evenness index (E). Three metabolic functional diversity indices were calculated as Equations (2)–(5). Functional diversity index was calculated using the AWCD value of the culture for 120 h.

(1) Shannon–Wiener diversity index (H′)

$$H' = -\sum P_i \ln P_i \tag{2}$$

$$P_i = (C_i - R)/\sum(C_i - R) \tag{3}$$

The P$i$ represents the ratio of the absorbance value in the $i$ th (1 to 31) well to the total absorbance values of all wells.

(2) Simpson diversity index (D)

$$D = 1 - \sum P_i^2 \tag{4}$$

(3) Shannon evenness index (E)

$$E = H'/\ln S \tag{5}$$

The S represents the total number of utilized carbon sources (31 carbon sources), the number of wells that vary in color.

### 2.6. Statistical Analysis

All experiments were performed in triplicate, and the results were expressed as the mean ± standard deviation. Excel 2016 and Origin 2018 were used for data analysis. The statistical significance of changes was determined by one-way analysis of variance (ANOVA) by SPSS 22.0. $p$ values of less than 0.05 were considered significant. The diversity indices, Shannon diversity (H′), Shannon evenness (E), and Simpson diversity (D) were analyzed using Duncan's multiple range test, separately.

## 3. Results and Discussion

### 3.1. Resuspension Events Test under Four Working Conditions

As shown in Figure 2a,b, under the four wind speeds, the average TN and TP concentrations in the overlying water gradually increased with the increase of the disturbance time, and finally tended to be stable. The resuspension event caused by the wind disturbance could promote the release of N and P from resuspended sediment into the water body, affecting the structure of the planktonic bacterial communities. In addition, with increased windspeed, this promotion effect was enhanced.

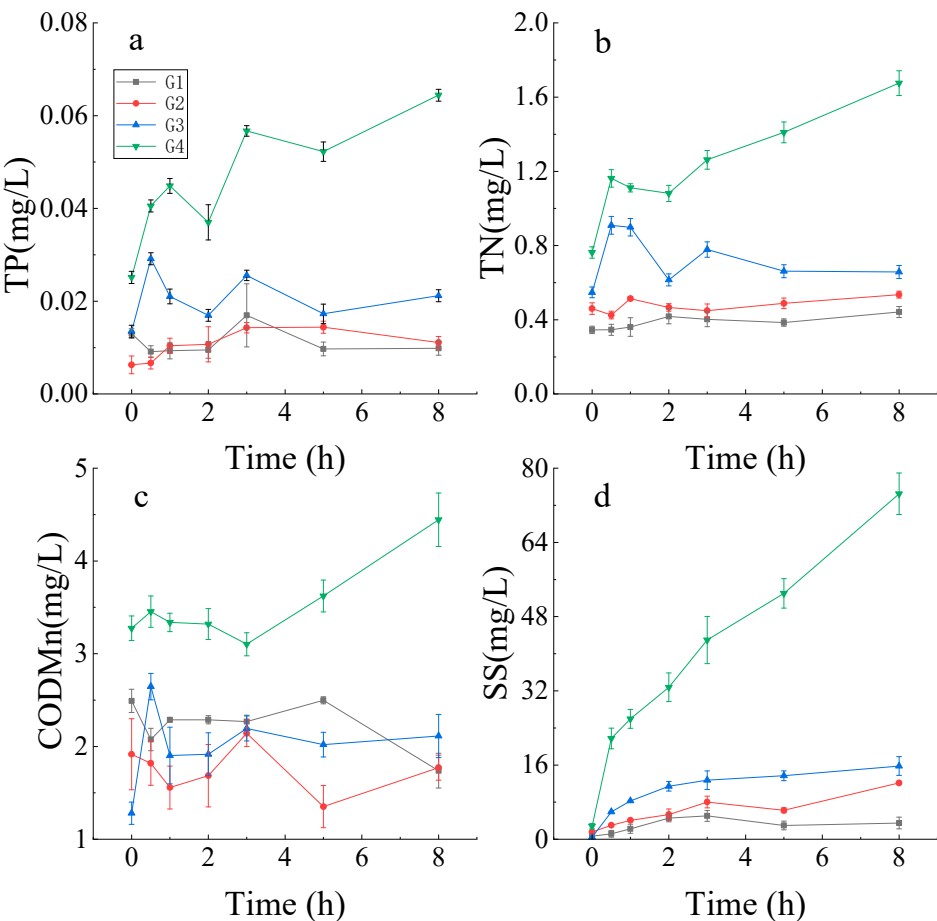

**Figure 2.** Changes in the average concentration of (**b**) total nitrogen (TN), (**a**) total phosphorus (TP), (**c**) chemical oxygen demand (COD), and (**d**) suspended solids (SS) of overlying water under four working conditions.

As shown in Figure 2c, in G1 and G2, there were significant differences between the COD concentrations of the G1 and G2 treatment groups during the whole test period ($p < 0.05$), and there was significant difference between the final value and initial value

of COD concentration in the G1 and G2 treatment groups ($p < 0.05$). In G3, the COD in the overlying water increased rapidly after 0.5 h of disturbance. Comparing to the initial concentration, the final growth rate was up to 73%. After reaching the peak, the concentration of COD kept fluctuating. In G4, the COD in the overlying water continued to increase, reaching a maximum of 4.42 mg/L after 8 h of disturbance, which was 65% higher than the initial concentration. In conclusion, the higher wind speed led to a stronger water disturbance, and then resulted in an increase in the rate of sediment suspension and a decrease in the sedimentation rate, thus, the concentration of COD in the overlying water was increased.

As shown in Figure 2d, with the increase of the disturbance time, the average SS concentration of the overlying water gradually increased under the four experimental conditions. Wind-induced wave mixing increased sediment resuspension and SS concentrations, with enhanced effects from higher velocity treatments (G3, G4). A similar conclusion was reported by Zheng et al. [55]. The results indicated that the disturbing effect of sediments became stronger with the increasing wind speed.

In order to study the influence of the dynamics of the sediment-overlying water system, the temperature of this experiment was maintained at a constant level. However, temperature is very important to carbon metabolism for its effects on some processes, such as the mineralization which causes the transformation of organic compounds into inorganic compounds in sediments with the participation of microorganisms, and which has been widely studied [56,57]. Grierson et al. found that increasing temperature increased specific P mineralization of soils from fertilized and unfertilized plantations of loblolly pine [58]. Trevathan-Tackett et al. found that under elevated seawater temperatures, carbon accumulation rates may diminish due to higher remineralization rates at the sediment surface, but the anoxic conditions ubiquitous to seagrass sediments can provide a degree of carbon protection under warming seawater temperatures [57].

### 3.2. The AWCD of All Carbon Sources in Sediment Microbial Communities within Incubation Time

Generally, the utilization of carbon sources was positively correlated with the metabolic capacity of the corresponding microorganisms, which could be characterized by AWCD [51]. At present, the Biolog profiles are generally referred as Community Level Physiological Profiles (CLPP) and have been widely used to explore the carbon metabolism and degradation activity of microorganisms [52,59–61]. The experiment was for 7 days, and the slope of each point of the AWCD curve represented the average metabolic rate of the microbial community (Table 1) [62]. Over 7 days, three periods of microbial activity could be divided, the adaptation period, logarithmic period, and stable period, which is consistent with a previous study [63]. It can be seen in Figure 3 that the AWCD of all samples within 0–24 h was very low, which indicated that the carbon sources in the BIOLOG ECO microplate were not used. In addition, the AWCD gradually increased with the incubation time, which was due to the extension of the culture time of the microorganisms. The metabolic activity was gradually enhanced, and reached the maximum (P) within 24–96 h, after 96 h the AWCD increased slowly and then stabilized (Figure 3). The carbon metabolism capacity of the sediment microbial community reached the maximum (K) in 168 h (7 d). Generally, it took an average of seven days, and at least five days, for the metabolic capacity of microorganisms to reach its maximum value, which is consistent with previous studies [32,64]. The average absorbance in the BIOLOG ECO microplate increased significantly, which indicated that the microbial communities in the four experimental conditions could use the carbon source in the BIOLOG ECO microplate during the stable period. Furthermore, the metabolic rate and AWCD were similar among the four sediment microbial communities ($p > 0.05$), which suggested that the resuspension events had no significant effect on the metabolic activity of the sediment microbial communities.

**Table 1.** The K and P of four sediment samples in four working conditions.

| Samples | K | P |
|---------|-------|-------|
| G1 | 1.72 | 0.039 |
| G2 | 1.717 | 0.035 |
| G3 | 1.769 | 0.035 |
| G4 | 1.718 | 0.026 |

K is the maximum value of the average well-color development (AWCD) and represents the utilization capacity of carbon sources, P is the maximum value of the slope of AWCD, and represents the maximum carbon metabolism rate.

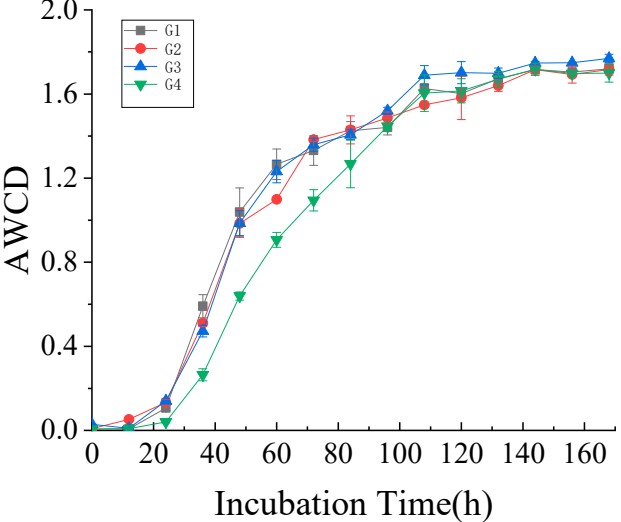

**Figure 3.** The AWCD of all carbon sources in sediment microbial communities within the incubation time.

### 3.3. AWCD of Different Biochemical Categories of Carbon Sources

According to their biochemical properties and molecular composition, the 31 carbon sources in the BIOLOG ECO microplates were divided into six categories, including polymers, phenolic acids, amines, amino acids, carboxylic acids, and carbohydrates. The details were described in Table 2. The average metabolic rate of the microbial community with six types of carbon source was calculated (Table 3). The results showed that the utilization capacity for polymer was the best, and the maximum carbon metabolism rate for carbohydrates was relatively low under the four working conditions. The AWCD values of the six different types were analyzed after the resuspension event experiment, and shown in Figure 4. The results indicated that with increasing cultivation time, the microbial utilization of the six kinds of carbon source showed an increasing trend. However, the microbial utilization rates of the six carbon sources were different in the four sediment microbial communities. Overall, the microorganisms had the lowest metabolic consumption for amines, and showed a preference for the other five carbon sources. Moreover, the highest AWCD value for polymer indicated that polymers were the carbon source with the highest metabolic utilization.

**Table 2.** The 31 kinds of carbon substrates in BIOLOG ECO microplates.

| Chemical Guild | Plate Number | Substrates | Chemical Formula |
|---|---|---|---|
| Polymers | C1 | Tween 40 | - |
| | D1 | Tween 80 | - |
| | E1 | α-Cyclodextrin | $C_{36}H_{60}O_{30}$ |
| | F1 | Glycogen | $(C_6H_{10}O_5)n$ |
| Carbohydrates | G1 | D-Cellobiose | $C_{12}H_{12}O_{11}$ |
| | H1 | α-D-Lactose | $C_{12}H_{12}O_{11}$ |
| | A2 | Methyl-D-glucoside | $C_7H_{14}O_6$ |
| | B2 | D-Xylose | $C_5H_{10}O_5$ |
| | C2 | i-Erythritol | $C_4H_{10}O_4$ |
| | D2 | D-Mannitol | $C_6H_{14}O_6$ |
| | E2 | N-Acetyl-D-glucosamine | $C_8H_{15}NO_6$ |
| | H2 | D,L-α-Glycerol phosphate | $C_3H_9O_6P$ |
| | B1 | Glucose-1-phosphate | $C_6H_{13}O_9P$ |
| | G2 | Pyruvic acid methyl ester | $C_4H_6O_3$ |
| Carboxylic acids | F2 | D-Glucosaminic acid | $C_6H_{13}NO_6$ |
| | A3 | D-Galactonic acid latone | $C_6H_{10}O_6$ |
| | B3 | D-Galacturonic acid | $C_6H_{10}O_7$ |
| | E3 | γ-Hydroxy butyric acid | $C_4H_8O_3$ |
| | F3 | Itaconic acid | $C_5H_6O_4$ |
| | G3 | α-Keto butyric acid | $C_4H_6O_3$ |
| | H3 | D-Malic acid | $C_4H_6O_5$ |
| Amino acids | A4 | L-Arginine | $C_4H_{14}N_4O_2$ |
| | B4 | L-Asparagine | $C_4H_8N_2O_3$ |
| | C4 | L-Phenylalanine | $C_9H_{11}NO_2$ |
| | D4 | L-Serine | $C_3H_7NO_3$ |
| | E4 | L-Threonine | $C_4H_9NO_3$ |
| | F4 | Glycyl-L-glutamic acid | $C_7H_{12}N_2O_5$ |
| Amines/amides | G4 | Phenylethylamine | $C_8H_{11}N$ |
| | H4 | Putrescine | $C_4H_{12}N_2$ |
| Phenolic acids | C3 | 2-Hydroxy benzoic acid | $C_7H_6O_3$ |
| | D3 | 4-Hydroxy benzoic acid | $C_7H_6O_3$ |

**Table 3.** The K and P of six types of carbon sources in four working conditions.

| Samples | Chemical Guild | K | P |
|---|---|---|---|
| G1 | Polymers | 2.228 | 0.118 |
| | Phenolic acids | 1.245 | 0.266 |
| | Amines | 0.578 | 0.156 |
| | Amino acids | 1.875 | 0.057 |
| | Carboxylic acids | 1.753 | 0.205 |
| | Carbohydrates | 1.312 | 0.101 |
| G2 | Polymers | 2.442 | 0.094 |
| | Phenolic acids | 1.273 | 0.084 |
| | Amines | 1.107 | 0.053 |
| | Amino acids | 1.868 | 0.067 |
| | Carboxylic acids | 1.425 | 0.069 |
| | Carbohydrates | 1.583 | 0.049 |

**Table 3.** *Cont.*

| Samples | Chemical Guild | K | P |
|---|---|---|---|
| G3 | Polymers | 2.453 | 0.086 |
| | Phenolic acids | 1.093 | 0.320 |
| | Amines | 0.383 | 0.211 |
| | Amino acids | 1.793 | 0.523 |
| | Carboxylic acids | 1.426 | 0.143 |
| | Carbohydrates | 1.711 | 0.055 |
| G4 | Polymers | 2.535 | 0.078 |
| | Phenolic acids | 1.070 | 0.186 |
| | Amines | 1.022 | 0.091 |
| | Amino acids | 1.704 | 0.075 |
| | Carboxylic acids | 1.686 | 0.065 |
| | Carbohydrates | 1.557 | 0.085 |

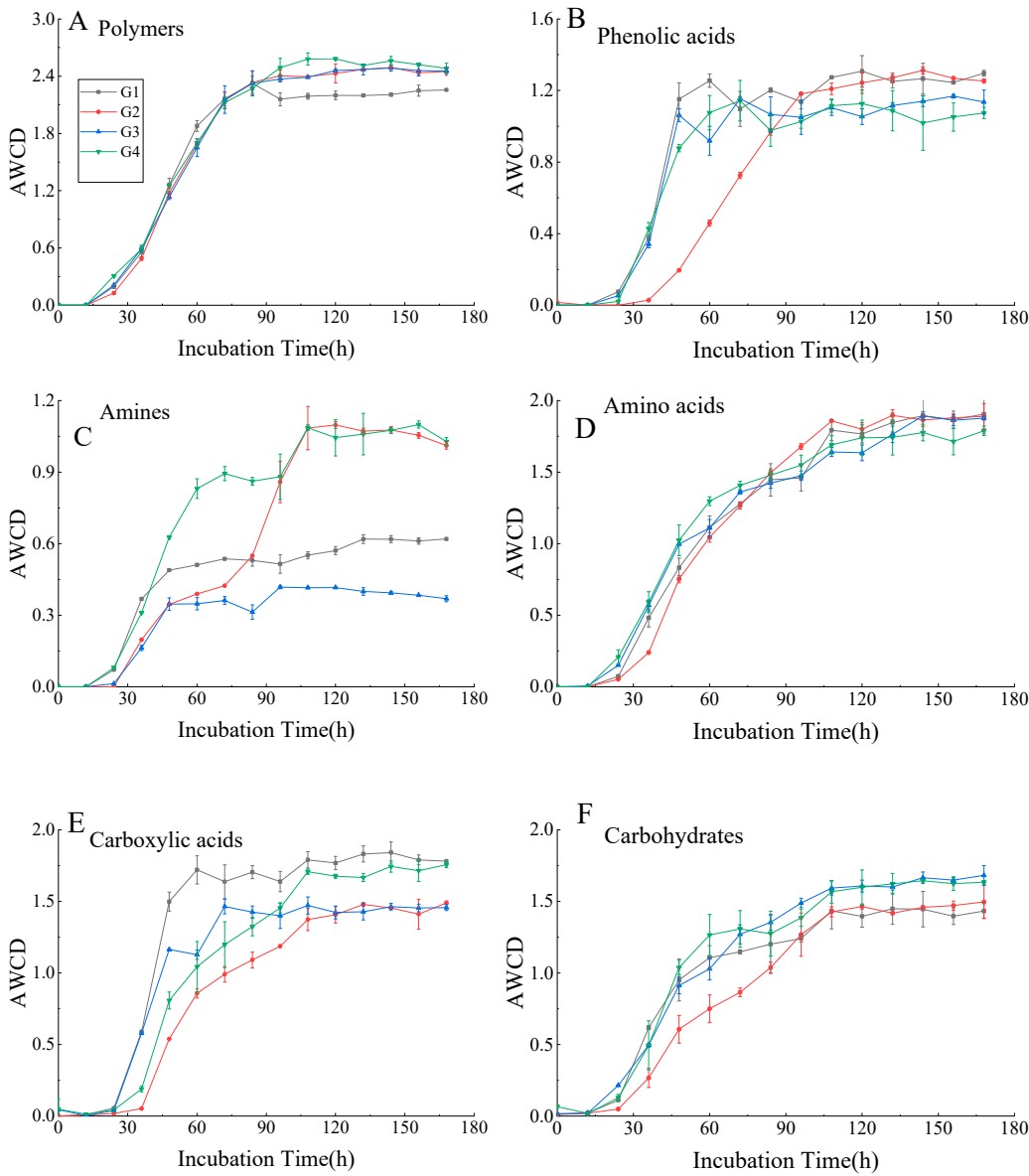

**Figure 4.** The AWCD of six types of carbon sources in the four sediment microbial communities, including polymers (**A**), phenolic acids (**B**), amines (**C**), amino acids (**D**), carboxylic acids (**E**), and carbohydrates (**F**).

For the type of carbohydrates, amino acids, polymers, and phenolic acids, there was no significant difference in the microbial utilization under the four different working conditions ($p > 0.05$). In contrast, significant difference was observed for amines and carboxylic acids ($p < 0.05$). The maximum value of AWCD (K) for amines in Table 2 represented the best microbial metabolism capacity. The significant difference in the utilization capacity and efficiency of amines can be observed in Figure 4C, and the utilization of amines required relatively longer periods of time, compared with other guilds [49], which needed to be studied further. In general, the utilization of amines was increased, while that of carboxylic acids was decreased as the wind–wave disturbance increased, which indicated that the resuspension events changed the preference of microbial carbon sources in sediments [32,65]. The reason for this phenomenon might be that the resuspension events changed the activity of the corresponding functional flora in the microbial community. Pusceddu et al. found that resuspension events increased the organic carbon availability for benthic consumers, which will affect the structure of the microbial community in the aquatic ecosystem [66]. Su et al. demonstrated that a change of environmental factors could cause a change of soil microbial community activity [67].

### 3.4. Microbial Metabolic Functional Diversity

The Shannon–Wiener diversity index and Simpson diversity index are linearly related to the relative abundance in different carbon source environments, while the Shannon uniformity index is only affected by uniformity.

It can be easily observed from Table 4 that there was no obvious difference of Shannon–Wiener diversity index, Simpson diversity index, and Shannon uniformity index between the control (G1) and disturbed groups (G2, G3, G4) ($p > 0.05$). However, significant difference was observed among the three disturbed groups. The Shannon–Wiener diversity index, Simpson diversity index, and Shannon uniformity index of the G2 group significantly differed from the other two disturbed treatment groups ($p < 0.05$). The Shannon diversity index and Simpson uniformity index of the G2 group were the highest, while that of the G4 group was the lowest, which indicated that the richness and uniformity of the sediment microbial species were enhanced under less disturbed conditions. The above results also showed that in the process of increasing disturbance conditions, three indexes had a trend of decreasing first and then increasing. The possible reason was that a weaker disturbance environment could promote the structure of microbial community to a certain extent, while a stronger disturbance reduced richness and uniformity of species.

**Table 4.** Comparison of metabolic functional diversity indices of the sediment microbial communities. Data in the table are expressed as mean ± standard deviation. H′ stands for Shannon–Wiener diversity index, D stands for Simpson diversity index, and E stands for Shannon evenness index.

| Sample | Shannon Diversity (H′) | Shannon Evenness (E) | Simpson Diversity (D) |
|---|---|---|---|
| G1 | 3.233 ± 0.034 | 0.941 ± 0.010 | 0.958 ± 0.002 |
| G2 | 3.249 ± 0.048 | 0.946 ± 0.014 | 0.959 ± 0.002 |
| G3 | 3.245 ± 0.041 | 0.941 ± 0.010 | 0.959 ± 0.001 |
| G4 | 3.166 ± 0.041 | 0.922 ± 0.011 | 0.956 ± 0.002 |

It should be noted that the present study was conducted in microcosms under controlled conditions of laboratory experiments. The extrapolation of results from any microcosm experiment should proceed with caution since the exposure conditions represented in the experiments are simplified compared to the field. In addition, further study is needed to investigate the responses of microbial carbon metabolism to sediment resuspension event in natural water.

## 4. Conclusions

In the present study, the process of sediment resuspension events and their potential effects on the microbial metabolic function in sediments were investigated. With the increase of wind–wave disturbance, the release of N and P from the resuspended sediment was promoted, which increased primary production and the risk of potential eutrophication. The total carbon metabolism in the sediment microbial communities, represented by AWCD, was maintained, while differences in the utilization capacity and rates of six different carbon sources indicated that short-term resuspension events will affect microbial utilization ability for some specific types of carbon sources. Therefore, these results suggested that sediment resuspension events may change the activity of some functional colonies, affecting the carbon cycle process of aquatic ecosystems. The AWCD of all carbon sources did not change significantly, which indicated that the short-term (8 h) resuspension events could not cause changes to the microbial community and total carbon metabolism.

Overall, the results showed differences in microbial metabolic pathways between amine and polymer utilization, and a higher microbial diversity under intermediate disturbance levels was observed. Our findings suggested that disturbed conditions could affect the metabolic function of sediment microorganisms. To fully understand the relationship between the disturbance conditions and the metabolic functions of sediment microorganisms, it is necessary to further study the mechanism of long-term resuspension events, and the effects of different disturbance conditions on the metabolic functions of microorganisms in the sediments.

**Author Contributions:** M.W.: Conceptualization, Methodology, Validation, Formal analysis, Resources, Data Curation, Writing—Original Draft, Writing—Review & Editing. M.Z.: Validation, Formal analysis, Validation, Resources, Data Curation, Writing—Original Draft, Visualization. W.D.: Method-ology, Validation, Formal analysis, Validation, Resources, Data Curation, Writing—Original Draft, Visualization. L.L.: Validation, Resources, Data Curation, Writing—Original Draft, Visualization. Z.L.: Validation, Resources, Data Curation, Writing—Original Draft, Visualization. L.M.: Conceptualization, Methodology, Investigation, Writing—Review & Editing, Supervision. J.H.: Conceptualization, Methodology, Writing—Review & Editing, Supervision. All authors have read and agreed to the published version of the manuscript.

**Funding:** Project supported by the National Key Plan for Research and Development of China (No. 2016YFC0401709), the National Science Funds for Creative Research Groups of China (No. 51421006), the National Natural Science Funds for Excellent Young Scholar (No. 51722902), the National Natural Science Foundation of China (No. 51979075, No. 51709081), and PAPD.

**Data Availability Statement:** The data presented in this study are available on request from the corresponding author. The data are not publicly available due to privacy.

**Conflicts of Interest:** The authors declare no conflict of interest. The funders had no role in the design of the study; in the collection, analyses, or interpretation of data; in the writing of the manuscript, or in the decision to publish the results.

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
