# Peer review of "Microbial Carbon Metabolic Functions in Sediments Influenced by Resuspension Event"

_water, doi:10.3390/w13010007_

Round 1

Reviewer 1 Report

The authors reported the responses of microbial carbon metabolism in sediments under different wind-wave disturbances that were analyzed by BIOLOG ECO microplates. This article shows interesting results and I recommend its publication in Water.

I suggest some aspects to be improved:

  • Please provide more details regarding the sediment samples collection. Did you use sterile sample boxes?
  • Do you have controls for all the methods used in this manuscript? It is very important to have references for comparison.
  • Please provide more details regarding the indoor annular flumes (Are they made in-house? How you manage to obtain these speed values? How does it work this device?)
  • The reference no. 35 is only for the gravimetric method of solids, or also for the rest of the methods. Please give references for the total nitrogen and total phosphorus and molybdenum antimony photometric method, and for potassium permanganate oxidation method.
  • Please check the reference no 39!!!
  • Please check the English language (ex. line 146).
  • There is no discussion regarding your results and the available literature.
  • Please provide the reason for selecting those time intervals describe in Figures 2, 3 and 4.
  • Please summarize the conclusions.

Author Response

Thanks for your constructive comments and suggestions. We have carefully revised the whole manuscript according to your kind suggestions one by one. Revised portion are marked in red in the paper. Please find the responses to the reviewer's detailed comments point by point in "Response to reviewers". 

Reviewer 2 Report

The aim of the paper is to provide more information regarding the potential impact of sediment resuspension events on the functional activities of microbial communities.

In situ Water Oxygen levels should be stated as well as the biological oxygen demand (BOD). Overall, although modest in the variety of assays performed, it is a well written paper.

  • 109 line, please cite method
  • 112 line,  regarding the Microplate Test Method, Biolog ecoplates is a relatively new method of determining the variety of the microbiota, the 31 substrates offer a quick and easy result, however, I would like to see stated the multifunctional enzyme label tester mentioned as in citation 40. Upon further check of citation 40 "Effects of Nanoplastics on Freshwater Biofilm Microbial Metabolic Functions as Determined by BIOLOG ECO Microplates" the multifunctional enzyme label tester was not mentioned as well.

Author Response

(The authors gave the same response as above.)

Reviewer 3 Report

The article is interesting and well prepared.

The discussion of the results with the scientific literature is very poor. It should be improved by adding more scientific news.

The authors presenting the diagrams 4 can calculate the kinetic values of microbial growth from the data. This data will be more interesting than the charts themselves. Count the specific speed of microorganisms, generation time, lag phase and log phase.

See the article: https://www.ncbi.nlm.nih.gov/pmc/articles/PMC98929/

This will make your work easier and enrich the article scientifically

The statistics in Table 3 are poorly done. There can be no homogeneous "ac" groups.

The authors did not write anything on temt. Among the basic parameters determining the success of the resuspension process, the key role is also assigned to temperature. As long as the mineralization processes
run the fastest in its mesophilic ranges, effective hygienisation is only ensured by its very high values. Under such conditions, the participation of microorganisms in the process becomes essential. Add more detail to my thought.

There is a thin layer in the sediments (from several millimeters to several centimeters), in which anaerobic processes dominate with the participation of bacteria. The breakdown of organic matter by bacteria is usually the primary mechanism for feeding the internal reservoir with nutrients. After exceeding the permissible load for a specific tank nutrients, the so-called process of supply or internal import, consisting in the release of nutrients, especially accumulated phosphates in bottom sediments. - Add it to your article

Author Response

(The authors gave the same response as above.)

Reviewer 4 Report

The subject of the publication is exciting and worth investigating. There is no simple model and method to approach the subject. In this manuscript, the authors assume that commercial technology (BIOLOG ECO) would be adequate to study the phenomenon of sediment resuspension.

In my opinion, there is no proof that this method reflects the actual biological process in natural conditions.

  1. What is the proof that the experimental conditions reelect processes occurring in natural conditions (windy conditions agitating lake sediment)? 
  2. Why is the title focusing on commercial technology and not the studied process? It looks a bit like an advertisement.
  3. The crucial impact would play dissolved oxygen and oxygen transfer rate. What are these values in natural conditions and microplate? They are entirely different from my experience, and the microplates do not appropriately reflect conditions found in the lake sediment.

Author Response

(The authors gave the same response as above.)

Round 2

Reviewer 1 Report

The authors addressed almost of the reviewers’ comments and the manuscript has improved. I recommend the publication of this paper in Water if the Editor approve it.

Author Response

Thanks for your recommendation. We sincerely hope this manuscript will be finally acceptable to be published in Water.

Reviewer 4 Report

The authors made efforts to clarify and correct the manuscript. However, I believe that the publication's quality will increase significantly through more concise, clear -cut, and less blurred descriptions.
For example, you can briefly list the limitations, advantages, and disadvantages of the model and how it differs from natural conditions. The fact that the experimental model does not fully reflect natural conditions is not a problem. Each model is an imperfect reflection of some biological process. However, a confusing description is a definite shortcoming, which does not fully define the experimental model, indirectly misleading that the observed changes can be directly related to natural conditions.
Another example of confusing description is the statement L64 "participation of methanogenic and methane-oxidizing bacteria and other microorganisms" is ambiguous. It could potentially suggest that methanogens are bacteria, which is not valid.
While quite complicated descriptions of results can be, to some extent, accepted in the results and discussions, the conclusions must be concise and clear-cut. Also, some statement does not reflect results. For example, L282:
"utilization of polymers and amines in 3.62 m/s group showed the highest and lowest rate, respectively".
The highest utilization for polymers was for G4-14.10 m/s (Table 2 and Fig 4A), not 3.62 m/s.
The lowest rate for amines in 3.62 m/s is the somehow questionable result. The effect is not dose-dependent and protruding. The amines utilization should be better discussed and maybe not suitable for clear-cut conclusions.

Author Response

Thanks for your constructive comments and suggestions. We have carefully revised the whole manuscript according to your kind advices and referees’ latest suggestions one by one. The minor revision has also been conducted using the “Track Changes” function in Microsoft Word and revised portion are marked in red in the paper. Please find the responses to the reviewer’s detailed comments point by point in “Response to reviewers”
